# Vaccine Based on Dendritic Cells Electroporated with an Exogenous Ovalbumin Protein and Pulsed with Invariant Natural Killer T Cell Ligands Effectively Induces Antigen-Specific Antitumor Immunity

**DOI:** 10.3390/cancers14010171

**Published:** 2021-12-30

**Authors:** Akihiro Watanabe, Kimihiro Yamashita, Mitsugu Fujita, Akira Arimoto, Masayasu Nishi, Shiki Takamura, Masafumi Saito, Kota Yamada, Kyosuke Agawa, Tomosuke Mukoyama, Masayuki Ando, Shingo Kanaji, Takeru Matsuda, Taro Oshikiri, Yoshihiro Kakeji

**Affiliations:** 1Division of Gastrointestinal Surgery, Department of Surgery, Graduate School of Medicine, Kobe University, Kobe 650-0017, Japan; wtnbnz@gmail.com (A.W.); kota.y.im.kobe@gmail.com (K.Y.); kyosukeagawa@gmail.com (K.A.); oicw23scofield@gmail.com (T.M.); hilugatafukurousan@gmail.com (M.A.); kanashin@med.kobe-u.ac.jp (S.K.); tmatsuda@med.kobe-u.ac.jp (T.M.); oshikiri@med.kobe-u.ac.jp (T.O.); kakeji@med.kobe-u.ac.jp (Y.K.); 2Center for Medical Education and Clinical Training, Kindai University Faculty of Medicine, 377-2 Onohigashi, Osaka 589-0014, Japan; mfujita47@gmail.com; 3Division of Gastrointestinal Surgery, Saiseikai Suita Hospital, Kawazono-cho, Suita 564-0013, Japan; akira.fine-day@hotmail.co.jp; 4Division of Gastrointestinal Surgery, Konan Medical Center, Kamokogahara, Higashinada, Kobe 658-0064, Japan; mnishi247@gmail.com; 5Department of Immunology, Kindai University Faculty of Medicine, 377-2 Ono-higashi, Osakasayama 589-0014, Japan; takamura@med.kindai.ac.jp; 6Department of Disaster and Emergency and Critical Care Medicine, Graduate School of Medicine, Kobe University, 7-5-2, Kusunoki-cho, Chuo-ku, Kobe 650-0017, Japan; masa9804chicco@gmail.com

**Keywords:** antitumor immunity, iNKT cell-enhanced vaccine, dendritic cells, electroporation, tissue-resident memory T cells

## Abstract

**Simple Summary:**

This study shows the potential of a novel dendritic cell vaccine therapy in antitumor immunity, in which bone marrow-derived dendritic cells are electroporated with an exogenous ovalbumin protein and simultaneously pulsed with α-galactosylceramide. This strategy enhances the induction of cytotoxic CD8^+^ T cells specific for tumor-associated antigens through the activation of invariant natural killer T cells, natural killer cells, and intrinsic dendritic cells. Moreover, this strategy sustains antigen-specific antitumor T cell responses over time.

**Abstract:**

(1) Background: Cancer vaccines are administered to induce cytotoxic CD8^+^ T cells (CTLs) specific for tumor antigens. Invariant natural killer T (iNKT) cells, the specific T cells activated by α-galactosylceramide (α-GalCer), play important roles in this process as they are involved in both innate and adaptive immunity. We developed a new cancer vaccine strategy in which dendritic cells (DCs) were loaded with an exogenous ovalbumin (OVA) protein by electroporation (EP) and pulsed with α-GalCer. (2) Methods: We generated bone marrow-derived DCs from C57BL/6 mice, loaded full-length ovalbumin proteins to the DCs by EP, and pulsed them with α-GalCer (OVA-EP-galDCs). The OVA-EP-galDCs were intravenously administered to C57BL/6 mice as a vaccine. We then investigated subsequent immune responses, such as the induction of iNKT cells, NK cells, intrinsic DCs, and OVA-specific CD8^+^ T cells, including tissue-resident memory T (T_RM_) cells. (3) Results: The OVA-EP-galDC vaccine efficiently rejected subcutaneous tumors in a manner primarily dependent on CD8^+^ T cells. In addition to the OVA-specific CD8^+^ T cells both in early and late phases, we observed the induction of antigen-specific T_RM_ cells in the skin. (4) Conclusions: The OVA-EP-galDC vaccine efficiently induced antigen-specific antitumor immunity, which was sustained over time, as shown by the T_RM_ cells.

## 1. Introduction

In recent years, remarkable progress has been made in cancer immunotherapy, especially with immune checkpoint inhibitors, such as for PD-1 inhibition [1,2]. Tumor-specific cytotoxic CD8^+^ T cells (CTLs) are the most important components for the optimal therapeutic efficacy of cancer immunotherapy [3]. Vaccines are the most common method to induce tumor-specific CTLs [4]. Various methods are applied for loading tumor antigen information to the hosts, including the use of tumor tissues as well as tumor-associated proteins, peptides, and nucleic acid information [5,6,7]. Among these, nucleic acid vaccines have recently been highlighted because they can efficiently induce the expression of antigen proteins in the hosts very efficiently in vivo, resulting in strong immune responses [8]. However, as nucleic acid vaccines require specific antigenic information, they are not always suitable for tumor vaccines for the following reasons. In the field of tumor vaccines, neoantigens (products of genetic mutations in tumor cells) are considered ideal target antigens because of their high immunogenicity [9]. However, it requires substantial efforts to specify the genetic information of neoantigens [9,10]. From this viewpoint, conventional protein-based vaccines still have the advantage in the field of tumor vaccines because they can make use of tumor lysates without specific identification of target antigens.

In vaccination, prime-boost immunization is usually applied to maximize immune responses, thereby activating antigen-specific CD8^+^ T cells in the hosts and subsequently inducing memory CD8^+^ T cells. In this process, CD8^+^ T cells require the cross-presentation of foreign antigens by antigen-presenting cells (APCs), such as dendritic cells (DCs) and immunological supports of helper CD4^+^ T cells, to achieve clonal expansion and long-term survival [11]. Recent studies have shown that invariant natural killer T cells (iNKT cells) enhance this process [12]. Here, iNKT cells have a distinct repertoire of T cell receptors (TCRs), with TCR α chains (Vα14–Jα18) are being paired with iNKT cell-specific TCR β chains (Vβ2, Vβ7, or Vβ8) in mice [13]. Consequently, they specifically recognize lipid antigens, such as α-galactosylceramide (α-GalCer), bound to the MHC-like molecule CD1d expressed on antigen-presenting cells. When α-GalCer systemically enters the host bodies together with protein/peptide antigens, iNKT cells interact with CD8α^+^ DCs. In turn, these DCs cross-prime antigens and recruit naive CD8^+^ T cells through a mechanism distinct from that of CD4^+^ helper T cells, enhancing antigen-specific CD8^+^ T cell responses [14]. Based on this concept, an “iNKT cell-enhanced vaccine” cancer immunotherapy strategy has been proposed, in which α-GalCer coexists with tumor antigens.

Among APCs, such as macrophages, B cells, and DCs that activate antigen-specific immune responses, DCs are the most potent APC population [15,16,17]. Previous studies have shown that the therapeutic administration of CD1d-expressing DCs pulsed with α-GalCer (forming α-GalCer-CD1d complex) inhibited tumor growth and metastasis and prolonged the survival of tumor-bearing hosts compared with free α-GalCer administration [18,19,20]. Mechanistically, multiple doses of free α-GalCer or the presentation of α-GalCer by B cells could induce NKT cell anergy, increase PD-1 expression, and decrease production of antitumor-related cytokines such as TNF, IL-2, or IFN-γ [21,22,23]. By contrast, when DCs present glycolipids such as α-GalCer, they also present specific co-stimulatory signals that strongly skew iNKT cells toward the Th1 phenotype [24]. In addition, administration of α-GalCer to CD8α^+^ DCs activates iNKT cells to induce their IFN-γ production [19,20]. These results indicate the possibility that DCs expressing the α-GalCer-CD1d complex can induce a strong immune response in the host.

Therefore, we hypothesized that the loading of exogenous proteins as antigens and α-GalCer pulsation on DCs might produce an efficient iNKT cell-enhanced vaccine. Using full-length ovalbumin (OVA) protein as a model antigen, we developed the “OVA-EP-galDC” system, in which bone marrow-derived immature dendritic cells (imDCs) were induced from mice and imDCs were electroporated with OVA, pulsed with α-GalCer, and stimulated with lipopolysaccharide (LPS) to induce maturation. After we confirmed the validity of the system, we evaluated its immunological/therapeutic efficacy for antitumor immune responses.

## 2. Materials and Methods

### 2.1. Mice

The procedures used in this experiment have been published previously [25]. Briefly, female C57BL/6J mice (8–12 weeks old) were purchased from CLEA Japan (Tokyo, Japan). C57BL/6-background CD45.1^+^ congenic mice (8–12 weeks old) were purchased from Jackson Laboratories (Bar Harbor, ME, USA). CD45.2^+^ OVA-specific OT-1 CD8^+^ TCR-transgenic mice (C57BL/6 background, recognizing the OVA peptide (SIINFEKL) mountable on H2Kb) were kindly provided by Dr. Akemi Sakamoto (Chiba University, Japan). All animal experiments were conducted in accordance with the institutional guidelines of Kobe University (approval number: P140802). All the mice were maintained under pathogen-free conditions and housed under a 12 h dark–light cycle at 22 ± 1 °C with ad libitum access to food and water. The mice were acclimatized for at least 5 days before the experiments.

### 2.2. Tumor Cell Lines

The procedures used in this experiment have been published previously [25]. Briefly, the EG7 cell line (a clone of EL4 murine thymoma cells expressing OVA) was purchased from American Type Culture Collection (Manassas, VA, USA). The B16-OVA murine melanoma cell line was kindly provided by Dr. Akemi Sakamoto (Chiba University, Chiba, Japan). The cell lines were maintained in RPMI-1640 medium supplemented with 10% (*v*/*v*) heat-inactivated fetal bovine serum (FBS), 2 mM glutamine, 100 U/mL penicillin, and 100 μg/mL streptomycin (Sigma-Aldrich, St Louis, MO, USA) at 37 °C in 5% CO_2_. The cultured cells tested negative for mycoplasma and viral contamination.

### 2.3. Antibodies, Flow Cytometry, and Peptide

The procedures used in this experiment have been published previously [25,26]. Immune cells were evaluated in each experiment and treated with FcR blocking reagent (Miltenyi Biotec, Bergisch Gladbach, Germany), stained with fluorescence-labeled antibodies as indicated, and incubated for 20 min at 4 °C in phosphate buffered saline (PBS) containing 0.1% bovine serum albumin. The following mouse antibodies were purchased from BioLegend (San Diego, CA, USA): anti-CD3 (labeled with FITC, PE, and APC-Cy7), anti-CD4 (PE), anti-CD8 (FITC and APC-Cy7), anti-CD11b (APC-Cy7), anti-CD11c (BV421), anti-CD40 (APC), anti-CD44 (PE), anti-CD45 (PerCP-Cy5.5), anti-CD45.1 (Pacific Blue), anti-CD45.2 (Pacific Blue), anti-CD62L (PerCP), anti-CD69 (PE), anti-CD80 (PerCP-Cy5.5), anti-CD86 (PE), anti-CD103 (PE-Cy7), anti-CD127 (BV510), and anti-NK1.1 (PerCP-Cy5.5.). We also used 7-AAD (BioLegend) to stain dead cells for the evaluation of cell viability. The stained cells were subjected to FACS Verse for data acquisition (Becton Dickinson, San Jose, CA, USA) and analyzed using FlowJo software (Tree Star, Ashland, OR, USA).

### 2.4. Immune Cell Isolation

The procedures used in this experiment have been published previously and applied with minor modifications [25,27,28]. Briefly, spleens were harvested from mice and mechanically minced to collect splenocytes. Next, the cell suspensions were filtered with cell strainers consisting of a nylon mesh with 70 μm pores (BD Biosciences, San Jose, CA, USA). Red blood cells were lysed with a lysis buffer (Sigma-Aldrich). To collect lymphocytes migrated in mouse skin, the skin was harvested from the mice, immersed in 0.25% tripsin and 2 mM EDTA, and incubated at 37 °C for 20 min. The epidermis was manually removed by forceps in 10 mL of 10% FBS-containing RPMI and mechanically minced. Next, the epidermis-derived cell suspensions were filtered through a 70 μm filter.

### 2.5. Tetramer Assay

The procedures used in this experiment have been published previously [29]. Briefly, the CD1d tetramer (MBL, Nagoya, Japan) was used to detect iNKT cells. First, α-GalCer (Tokyo Chemical Industry, Tokyo, Japan) was dissolved in pyridine to a concentration of 1 mg/mL and further diluted to a final concentration of 200 μg/mL in 0.9% NaCl/0.5% Tween-20. Next, 5 μL of the α-GalCer solution was further dissolved in 100 μL of the CD1d tetramer solution provided in the kit and incubated at 37 °C for 12 h to load. The α-GalCer-loaded CD1d tetramer solution was diluted 50 times, and the cells were stained with the diluted CD1d tetramer solution and the anti-CD4 antibody.

The H-2Kb/OVA (SIINFEKL)-tetramer (MBL, Nagoya, Japan) was used to detect OVA-specific CD8^+^ T cells. Immune cells in the spleen or skin were harvested from the mice, as described in Section 2.3, and stained with the H-2Kb/OVA-tetramer and the FITC-labeled anti-mouse CD8 antibody. The stained cells were then subjected to a flow cytometry analysis.

### 2.6. Preparation of Bone Marrow-Derived Immature Dendritic Cells (imDCs)

The procedures used in this experiment have been published previously [30]. Briefly, bone marrow cells were harvested from the femur and tibia of wild-type mice and cultured for 10 days in the presence of 10 ng/mL murine GM-CSF (Peprotech, East Windsor, NJ, USA). The culture medium was replaced every 2 days, and GM-CSF was also replenished. Non-adherent and weakly adherent cells were collected as the imDCs.

### 2.7. Electroporation and Coculture of imDCs with Proteins

The procedures used in this experiment have been published previously and applied with minor modifications [31]. Briefly, the GT Transfection System (MaxCyte, Gaithersburg, MD, USA) was used to load proteins into the imDCs by electroporation. The imDCs were resuspended in the 400 μL of the electroporation buffer provided in the kit at 1 × 10^8^ cells/mL and mixed with either 100 μg/mL of non-labeled OVA or 100 μg/mL of Alexa Fluor 488-labeled OVA (Thermo Fisher Scientific, Waltham, MA, USA). Electroporation was then conducted with the imDC suspensions using a preoptimized device setting (Mouse DC1 program). The electroporated imDCs were transferred to ULA T75 flasks and incubated for 2 h at 37 °C in 5.0% CO_2_. In parallel, some fresh imDCs were resuspended to 5 × 10^5^ cells/mL and cocultured for 2 h with either non-labeled OVA or Alexa Fluor 488-labeled OVA at a final concentration of 20 μg/mL.

### 2.8. Microscopic Evaluation of imDCs

The procedures used in this experiment have been published previously and applied with minor modifications [32]. We used an Olympus IX71 inverted fluorescence microscope (Olympus, Tokyo, Japan) to observe protein-loaded imDCs in bright field and fluorescence settings.

### 2.9. α-GalCer Pulsation on imDCs

The procedures used in this experiment have been published previously [30]. Briefly, some imDCs (non-treated cells or those electroporated with OVA) were cultured with 100 ng/mL of α-GalCer (KRN7000: Funakoshi, Tokyo, Japan) for 40 h before cell harvesting.

### 2.10. LPS Stimulation of imDCs for Maturation

The procedures used in this experiment have been published previously [30]. Briefly, some imDCs (those electroporated with OVA, those pulsed with α-GalCer, or treated with both) were cultured with 100 ng/mL of LPS (Wako Pure Chemicals, Osaka, Japan) for 16 h before cell harvesting for maturation. The LPS-stimulated cells are abbreviated as follows: “OVA-EP-DCs” for those electroporated with OVA alone, “galDCs” for those pulsed with α-GalCer alone, and “OVA-EP-galDCs” for those treated with both.

### 2.11. Preparation of OT-1 CD8^+^ T Cells

The procedures used in this experiment have been published previously [33]. Briefly, splenocytes were harvested from CD45.2^+^ OT-1 mice, and CD8^+^ T cells were purified using the MACS CD8^+^ T Cell Isolation Kit (Miltenyi Biotec, Bergisch Gladbach, Germany), according to the manufacturer’s instructions.

### 2.12. In Vitro OT-1 Proliferation Assay

The procedures used in this experiment have been published previously and were applied in this study with minor modifications [34]. Briefly, OT-1 CD8^+^ T cells were stained with CFSE using the Cell Trace™ CFSE Cell Proliferation Kit (Thermo Fisher Scientific, Waltham, MA, USA), according to the manufacturer’s instructions. CFSE-labeled OT-1 CD8^+^ T cells (1.0 × 10^6^ cells) were cocultured with LPS-stimulated control DCs or OVA-EP-DCs at 3.3 × 10^5^ cells in 24 well plates. The cell mixture was incubated for 5 days at 37 °C under 5% CO_2_ and then analyzed by flow cytometry.

### 2.13. DC-Based Immunization of Mice

The procedures used in this experiment have been published previously and were applied in this study with minor modifications [30]. Briefly, naïve wild-type mice received intravenous injections of one of the following: PBS, OVA-EP-DCs, galDCs, or OVA-EP-galDCs. Each mouse received 5 × 10^5^ DCs through the tail vein. The mice were sacrificed on days 4 and 7 to harvest the splenocytes for flow cytometry.

### 2.14. Pretreatment with OT-1

The procedures used in this experiment have been published previously [33]. Briefly, splenocytes were harvested from CD45.2^+^ OT-1 mice, and CD8^+^ T cells were purified by the MACS CD8^+^ T Cell Isolation Kit (Miltenyi Biotec, Bergisch Gladbach, Germany). Purified CD45.2^+^ OT-1 CD8^+^ T cells were intravenously transferred into C57BL/6-background CD45.1^+^ congenic mice through the tail vein. Each mouse received 5 × 10^5^ purified OT-1 CD8^+^ T cells. On the following day, the mice further received intravenous injections of DCs, as described above.

### 2.15. Subcutaneous Tumor Model

The procedures used in this experiment have been published previously and were applied in this study with minor modifications [35]. Briefly, 14 days following the DC immunization described in Section 2.13 (set as day 0), the mice subcutaneously received 2 × 10^5^ EG7 in the back. In addition, some mice further received the following antibodies intraperitoneally on days −3, −1, 2, and 5, and every 7 days thereafter until day 28: mouse anti-CD8a monoclonal antibody (clone 2.43: Bio X Cell, Lebanon, NH, USA) at 150 μg/mouse for CD8^+^ T cell depletion, rabbit anti-asialo GM1 antibody (FUJIFILM Wako Pure Chemical Corporation, Tokyo, Japan) at 100 μL/mouse for NK cell depletion, and isotype control (clone LTF-2: Bio X Cell, Lebanon, NH, USA) at 150 μg/mouse. Flow cytometry confirmed that 97% of the CD8^+^ T cells and 99% of the NK cells in the peripheral blood were depleted.

The tumor size was measured twice in a week using a caliper. The tumor volume was calculated using the following formula: V = 1/2 (L × W^2^), where L = length and W = width. The mice were euthanized when the maximum tumor diameter exceeded 20 mm, or their body weight loss exceeded 20%.

### 2.16. Lung Metastasis Tumor Model

The procedures used in this experiment have been published previously [36]. Briefly, naïve wild-type mice received 5 × 10^5^ B16-OVA cells intravenously through the tail vein. On day 3 after the tumor challenge, the mice received OVA-EP-galDCs, galDCs, or PBS as DC-based vaccines as described in Section 2.13.

### 2.17. IFN-γ ELISPOT Assay

The procedures used in this experiment have been published previously [37]. Briefly, CD8^+^ T cells were purified using the MACS CD8^+^ T Cell Isolation Kit (Miltenyi Biotec), according to the manufacturer’s instructions. The purified CD8^+^ T cells were cocultured with the DCs pulsed with the OVA peptide (SIINFEKL) for 20 h. An IFN-γ ELISPOT assay was then conducted according to the manufacturer’s instructions (BD Biosciences). The number of spots was manually counted.

### 2.18. Statistical Analysis

To analyze statistical differences among multiple groups, one-way analysis of variance (ANOVA) with Holm’s post hoc test was performed. The Student’s *t*-test was performed to analyze differences between two groups. The Kaplan–Meier test was performed to evaluate survival. These statistical analyses were conducted using JMP software (SAS Institute Inc., Cary, NC, USA). The data are presented as dot plots, unless specifically stated. Statistical significance was set at *p* < 0.05.

## 3. Results

### 3.1. Dendritic Cells Loaded with Ovalbumin Protein by Electroporation (OVA-EP-DCs) Effectively Stimulated OT-1 CD8^+^ T Cells

To produce efficient iNKT cell-enhanced vaccines, we first electroporated the full-length ovalbumin (OVA) protein into fresh bone marrow-derived imDCs. The non-treated imDCs and the imDCs electroporated with non-labeled OVA microscopically showed no fluorescence signal (Figure 1A). By contrast, we observed fluorescence signals in the imDCs cocultured with Alexa Fluor 488-labeled OVA and the imDCs electroporated with Alexa-Fluor 488-labeled OVA. The imDCs electroporated with Alexa Fluor 488-labeled OVA showed stronger fluorescence intensity than those cocultured with Alexa Fluor 488-labeled OVA (Figure 1A). The fluorescence signals of the imDCs were quantified by flow cytometry (Figure 1B). Consistent with the microscopical findings (Figure 1A), the imDCs electroporated with Alexa Fluor 488-labeled OVA showed higher signals than the imDCs cocultured with Alexa Fluor 488-labeled OVA (*p* < 0.0001; Figure 1B). We observed no fluorescence signal in the non-treated imDCs and the imDCs loaded with non-labeled OVA. In parallel, various concentrations of OVA were tested in a dose-escalation manner to determine the appropriate OVA concentration for coculture. The imDCs cocultured with 100 μg/mL of OVA produced almost identical signal intensities to the imDCs electroporated with the same concentration of OVA but died at higher frequencies (more than 80%). To maximize the dose of OVA to administer as well as to minimize the treatment-induced cell death, we chose OVA concentrations of 20 μg/mL for coculturing and of 100 μg/mL for electroporation.

As electroporation is cytotoxic, we evaluated the viability of the imDCs that underwent electroporation. The viability of the electroporated imDCs was lower than that of the non-treated imDCs (*p* = 0.0054, Figure 1C) but showed no significance compared with that of the imDCs cocultured with OVA (*p* = 0.0531, Figure 1C). These data suggest that the electroporation to imDCs were conducible with their minimum loss of the cells.

Next, we used the OT-1 transgenic mouse system to evaluate the efficiency of the protein-loaded DCs for antigen-specific CD8^+^ T cell responses. In these experiments, the DCs were stimulated with LPS for maturation and cocultured with OT-1 mouse-derived naïve CFSE-labeled CD8^+^ T cells. As a result, the OVA-EP-DCs remarkably accelerated the proliferation of OT-1 CD8^+^ T cells compared with the non-treated DCs (*p* = 0.00016; Figure 1D). These data suggest that electroporation of antigen proteins into DCs could efficiently stimulate antigen-specific CD8^+^ T cells in vitro.

### 3.2. OVA-EP-galDC Vaccine Promoted Complete Rejection of Subcutaneous Tumors in a Manner Dependent on CD8^+^ T Cells in Prophylactic Models

Based on the above findings, we attempted to verify the antitumor effects of the OVA-EP-galDC vaccine in vivo. For this purpose, the mice were preimmunized with PBS, the galDC, or the OVA-EP-galDC and then challenged with the EG7 cell line expressing OVA as a tumor antigen (Figure 2A). The OVA-EP-galDC vaccine significantly inhibited tumor growth compared with the PBS (*p* = 0.0390) or the galDCs (*p* = 0.0444). Next, to evaluate the direct impact of CD8^+^ T cells and NK cells on inhibiting the tumor growth, we depleted each of these cell types using specific antibodies (Figure 2B). The depletion of CD8^+^ T cells significantly reduced the antitumor effects of the OVA-EP-galDC (*p* < 0.0001 against the OVA-EP-galDC + isotype control, *p* = 0.8013 against PBS + isotype control; Figure 2B, blue line). The depletion of NK cells decelerated the tumor growth to some extent but did not achieve complete rejection of the tumor (*p* < 0.0001 against the OVA-EP-galDCs + isotype control; Figure 2B, black dot-dash line with diamond). These data suggest that the OVA-EP-galDC vaccine promoted the complete rejection of subcutaneous tumors in a manner primarily dependent on CD8^+^ T cells and NK cells to a lesser extent to in prophylactic models.

### 3.3. OVA-EP-galDC Vaccine Prolonged the Survival of Mice Bearing Lung Metastases

To extend the evaluation of tumor resistance by the OVA-EP-galDC vaccine, we conducted a survival study of lung metastasis model using the B16-OVA murine melanoma cells (Appendix A). As a result, the OVA-EP-galDC vaccine prolonged survival of the tumor-bearing mice compared with the PBS treatment (Appendix A).

### 3.4. OVA-EP-galDC Vaccine Induced iNKT Cells, NK Cells, and DCs in the Spleen

As immunological tumor elimination was shown in response to the OVA-EP-galDC vaccine, we next sought to evaluate the immune response cascades (iNKT/NK cells, DCs, and T cells) elicited in the hosts. For this purpose, we examined the splenocytes of the DC-immunized mice on day 4 by flow cytometry (Figure 3A). First, iNKT cells and NK cells were examined. The galDC vaccine or the OVA-EP-galDC vaccine significantly increased the frequency and the absolute number of iNKT cells and NK cells in the spleens of the immunized mice compared with those immunized with PBS or the OVA-EP-DC (Figure 3B and Appendix A). Consistent results were observed on day 7 (Appendix A). Next, we addressed whether the DC vaccine would further activate intrinsic DCs in the hosts. The galDC vaccine or the OVA-EP-galDC vaccine significantly increased the frequency and the absolute number of CD11c^+^ CD86^+^-activated DCs (Figure 3C and Appendix A) as well as CD40^+^ and CD80^+^-activated DCs (Appendix A) in the spleens of the immunized mice compared with those immunized with PBS or the OVA-EP-DCs. These data suggest that the α-GalCer pulsation on DCs efficiently activates iNKT cells, NK cells, and intrinsic DCs in hosts immunized with DCs.

Once DCs become activated, they interact with CD4^+^ and CD8^+^ T cells, the latter of which elicits direct antitumor activities. Therefore, we next evaluated the status of antigen-specific CD8^+^ T cells in the spleen on day 7 after the DC immunization (Figure 3D). The galDC vaccine or the OVA-EP-galDC vaccine significantly increased the absolute number of splenocytes (*p* < 0.0001 against galDCs, *p* < 0.0001 against OVA-EP-galDCs; Figure 3E, left graph), the frequency of H-2Kb/OVA-tetramer^+^ CD8^+^ T cells (*p* = 0.0042 against galDCs, *p* = 0.0048 against OVA-EP-galDCs; center graph), and the absolute number of H-2Kb/OVA-tetramer^+^ CD8^+^ T cells (*p* = 0.0020 against galDCs, *p* = 0.0027 against OVA-EP-galDCs; right graph) compared with PBS. These data suggest that the OVA-EP-galDC vaccine effectively induces iNKT cells, NK cells, and DCs in the spleen.

### 3.5. OVA-EP-galDC Vaccine Activated OT-1 CD8^+^ Effector T Cells in the Spleen

As we did not clearly observe antigen-specific CD8^+^ T cell responses with the DC vaccines alone (Figure 3D,E), we decided to enhance antigen-responsible naïve T cells in the hosts using the CD45.2^+^ OT-1 mouse system. Here, we transferred OT-1 CD45.2^+^ CD8^+^ T cells into CD45.1^+^ congenic mice on day −1 and immunized the host mice on day 0 with one of the followings PBS, OVA-EP-DCs, galDCs, or OVA-EP-galDCs (Figure 4A). The OVA-EP-galDC vaccine significantly increased the frequency and the absolute number of OVA-specific CD8^+^ T cells compared with the others (Figure 4B). We also evaluated CD8^+^ CD44^+^ CD62L^-^ KLRG1^-^ CD127^+^ memory precursor T cells. The OVA-EP-galDC vaccine significantly increased the frequency and the absolute number of the memory precursor T cells compared with others (Figure 4C). These data clearly suggest that, when the hosts were boosted with OT-1 OVA-specific CD8^+^ T cells prior to the vaccine, the OVA-EP-galDC vaccine effectively activated the OVA-specific T cells in the host spleen.

### 3.6. OVA-EP-galDC Vaccine Induced OT-1 CD8^+^ Memory T Cells in the Spleen

As we observed the induction of the memory precursor T cells in response to the OVA-EP-galDC vaccine, we directed our focus to the induction of long-term memory CD8^+^ T cells (Figure 5A) using the OT-1 congenic system described above. When we evaluated OT-1-derived CD45.2^+^ CD8^+^ T cells reactive for H-2Kb/OVA-tetramer in the spleens of the immunized mice on day 50, the OVA-EP-galDC vaccine induced the H-2Kb/OVA-tetramer reactive CD8^+^ T cells most efficiently (Figure 5B). Consistent with these data, the ELISPOT assay revealed that the OVA-EP-galDC vaccine promoted IFN-γ production by these OVA-specific CD8^+^ T cells most efficiently (Figure 5C). These data suggest that the OVA-EP-galDC vaccine efficiently induces OVA-specific long-term memory T cells in the spleen.

### 3.7. OVA-EP-galDC Vaccine Induced OT-1 CD8^+^ Tissue-Resident Memory T Cells (T_RM_ Cells) in the Skin

Recent studies have shown that tissue-resident memory T cells (T_RM_ cells) in the skin plays important roles in the rejection of subcutaneous tumors [38,39]. As we observed the complete rejection of subcutaneous tumors in response to the OVA-EP-galDC vaccine (Figure 2A,B), we sought OVA-specific T_RM_ cells in the skin. For this purpose, immune cells in the skin were examined using the OT-1 congenic system described above (Figure 6A). First, we evaluated OT-1-derived CD45.2^+^ CD8^+^ T cells reactive for H-2Kb/OVA-tetramer in the skin of the immunized mice on day 14. Whereas the OVA-EP-galDC vaccine did not induce non-specific CD8^+^ T cells in the skin (Figure 6B), we observed increases in OVA-specific OT-1 CD8^+^ T cells (H-2Kb/OVA-tetrameter^+^ CD45.2^+^ CD3^+^ CD8^+^ T cells) (Figure 6C) and OT-1-derived CD8^+^ T_RM_ cells (H-2Kb/OVA-tetrameter^+^ CD45.2^+^ CD3^+^ CD8^+^ CD69^+^ CD103^+^ T cells) (Figure 6D). In summary, our data suggest that the OVA-EP-galDC vaccine efficiently induces antigen-specific antitumor immunity and that this is sustained over time, as shown by the kinetics of the T_RM_ cells.

## 4. Discussion

Our data clearly demonstrated that the vaccine based on DCs electroporated with full-length OVA protein and pulsed with α-GalCer (OVA-EP-galDCs) effectively induced antigen-specific antitumor immunity to achieve complete tumor rejection in the prophylactic model. The OVA-EP-galDC vaccine induced iNKT cells, NK cells, and intrinsic DCs in the spleen compared to the galDCs. In the mice transferred with OT-1 CD8^+^ T cells, the OVA-EP-galDC vaccine effectively induced OT-1 CD8^+^ effector T cells and long-lasting OT-1 CD8^+^ memory T cells in the spleen. Consistently, the OVA-EP-galDC vaccine induced OT-1 CD8^+^ T_RM_ cells in the skin, which was speculated to contribute to the complete rejection of subcutaneous tumors in a prophylactic manner. Based on these findings, we concluded that the OVA-EP-galDCs represent an efficient iNKT cell-enhanced vaccine.

We successfully developed the OVA-EP-galDC vaccine system by electroporating full-length OVA protein to bone marrow-derived imDCs followed by the pulsation of α-GalCer (Figure 1). The advantage of using full-length protein antigens for the vaccine is that it allows us to use the lysates of patient-derived tumor tissues. It is generally very difficult to specify the information of tumor antigens [9,10]. Nevertheless, the multiple targeting of tumor antigens is indispensable to control and inhibit tumor growth in vivo. Therefore, our strategy might be able to induce antigen-specific antitumor immune responses by targeting a wide range of tumor antigens without specifying their information. To propose the antigen-EP-galDC system as a feasible strategy of personalized antitumor vaccine, we are currently attempting to apply tumor cell lysates to this system.

However, the OVA-EP-galDC vaccine did not increase the H-2Kb/OVA-tetramer^+^ CD8^+^ T cells (Figure 3E); in addition, the prophylactic transfer of naïve OT-1 CD8^+^ T cells prior to the DC vaccines barely caused slight increases (Figure 4). We speculate two reasons for the lack of marked increases in the H-2Kb/OVA-tetramer^+^ CD8^+^ T cells. One is that our single-tetramer experimental system did not allow us to detect all tumor-reactive T cells. That is, we could not examine OVA-reactive T cells that were indetectable by the H-2Kb/OVA-tetramer. To overcome this issue, we need other methods to analyze tumor-reactive T cells comprehensively, such as T cell receptor repertoire analysis. Another reason is that the adjuvant effects of α-GalCer may have been insufficient. To overcome this issue, the use of novel iNKT cell glycolipid ligands with more potent adjuvant effects than α-GalCer should be considered [40].

Followed by the inductions of the activated effector T cells (Figure 4) and long-lasting memory T cells (Figure 5), the OVA-EP-galDC vaccine induced the OVA-specific T_RM_ cells in the skin of the DC-immunized mice (Figure 6D). It is essential to achieve the proper activation and maintenance of tumor-specific CD8^+^ T cells for effective tumor growth inhibition [41]. In this regard, memory CD8^+^ T cells are classified into two major subpopulations. One is central memory T (T_CM_) cells, or effector memory T (T_EM_) cells, which recirculate in the peripheral blood throughout the body. The other is tissue-resident memory T (T_RM_) cells, which consistently reside in the skin, lung, liver, and gastrointestinal tracts [42,43,44,45]. T_RM_ cells rapidly respond to exert antigen-specific immunological protection once they encounter local antigens [46]. In addition, the presence of T_RM_ cells in tumor sites has been shown to correlate with a favorable prognosis of cancer-bearing hosts [47,48,49]. Furthermore, a recent report has shown that the additional use of α-GalCer to peptide-based vaccines against a malaria infection enhances the accumulation of T_RM_ cells in the liver to afford better protection [50], which supports the validity of our results. Since it is important to induce T_RM_ cells strongly for potent antitumor immunity [51], our iNKT cell-enhanced vaccine represents an appropriate cancer vaccine strategy that can induce antigen-specific T_RM_ cells.

Although the OVA-EP-galDC vaccine successfully rejected subcutaneous tumors, the galDC vaccine alone had little effect on tumor growth (Figure 2A), even with the activation of NK cells in both vaccines (Figure 3C). As these results suggest that that NK cells in this model exerted little effector functions, we needed to re-evaluate in detail whether the galDC vaccine system (including the OVA-EP-galDCs) could induce the effector functions of NK cells. As the effector function of NK cells has been shown to be important in the control of metastatic lung tumors independent of CD4^+^ and CD8^+^ T cells [52,53], we tested our DC vaccine system in a mouse lung metastasis model (Appendix A). As a result, the galDC vaccine alone controlled the growth of lung metastases in mice well, suggesting the antitumor effector functions of NK cells. In addition, NK cell depletion by the anti-asialo GM1 antibody after the OVA-EP-galDC vaccine reduced the therapeutic effects (Figure 2B). These results suggest that NK cells play at least a supportive/modulative role in the effector functions of antigen-specific CD8^+^ T cells. Taken together, these data indicate that, although NK cells alone are insufficient to inhibit the growth of certain tumor types (such as subcutaneous tumors), they exert immunomodulatory functions in our system. To better understand these iNKT/NK cell functions in antitumor immunity, we should reevaluate cell-to-cell-basis interaction among iNKT cells, NK cells, DCs, and T cells in detail.

The combination therapy of α-GalCer and immune checkpoint inhibitors (ICIs) has been shown to have synergistic effects [54]. To improve the therapeutic effects of the OVA-EP-galDCs, we are currently attempting to combine this system with ICIs as not only a prophylactic strategy but a therapeutic strategy for established tumors. It is also important to note that ICIs enhance the effector functions of antigen-specific T cells. In this study, however, we did not fully evaluate the therapeutic effects of the OVA-EP-DCs that are expected to induce antigen-specific T cells (Figure 2) because we primarily focused on the significance of α-GalCer-mediated immune responses in the EP-DC system. In this regard, the OVA-EP-DCs were insufficient to induce antigen-specific skin T_RM_ cells, which reflect the therapeutic effects (Figure 3D). Therefore, the effects of the OVA-EP-DCs are presumed to be weak. Nevertheless, particularly in the established tumor models, the OVA-EP-DCs (as well as the OVA-EP-galDCs) are expected to be effective in combination with ICIs, and this point should be verified in the future.

## 5. Conclusions

DC vaccines with the electroporation of full-length ovalbumin (OVA) protein, followed by the pulsation of α-GalCer, an iNKT cell ligand, effectively induce sustained antigen-specific antitumor immunity through the activation of iNKT cells and antigen-specific CD8^+^ T cells.

## Figures and Tables

**Figure 1 cancers-14-00171-f001:**
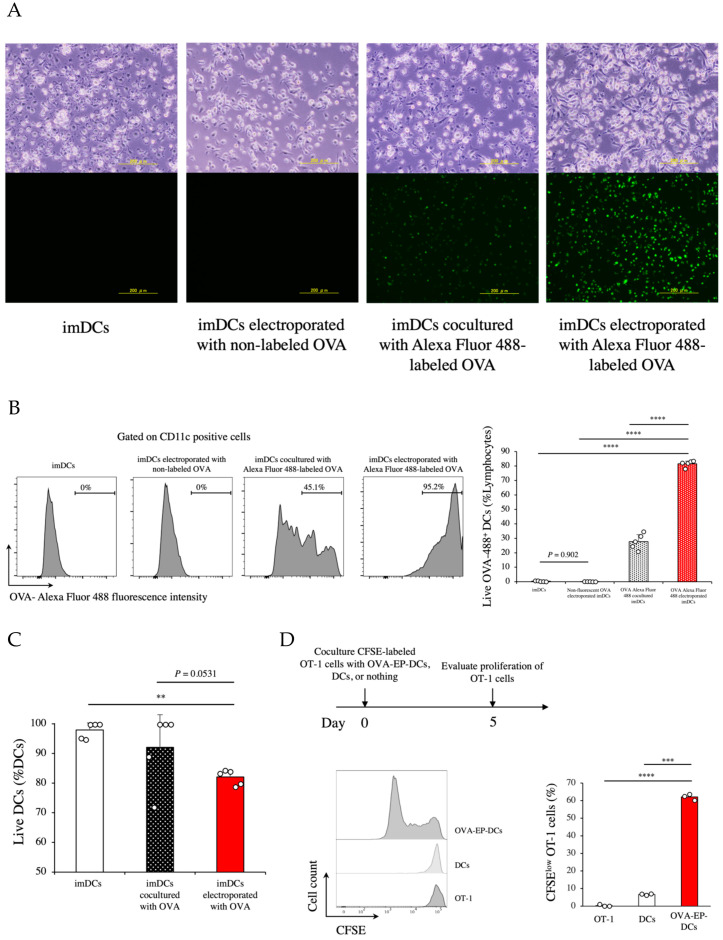
Dendritic cells loaded with ovalbumin protein by electroporation (OVA-EP-DCs) effectively stimulated OT-1 CD8^+^ T cells. Immature dendritic cells (imDCs) derived from murine bone marrow were electroporated with full-length ovalbumin (OVA) protein using the program of MaxCyte System. (**A**) The photomicrographs show control imDCs, imDCs electroporated with non-labeled OVA, imDCs cocultured with Alexa Fluor 488 labeled OVA, or imDCs electroporated with Alexa Fluor 488 labeled OVA. (**B**) Histograms represent the frequency of OVA loading into the imDC. The Alexa Fluor 488 fluorescence intensity was measured by flow cytometry in imDCs, imDCs electroporated with non-labeled OVA, imDCs cocultured with Alexa Fluor 488 labeled OVA, or imDCs electroporated with Alexa Fluor 488 labeled OVA. The graph shows the quantification of the frequency of OVA loading into the imDCs in each group (*n* = 5). (**C**) The viability of imDCs electroporated with OVA was compared with imDCs cocultured with OVA and control imDCs (*n* = 5). (**D**) Proliferation rates of OT-1 CD8^+^ T cells cultured alone, or cocultured with DCs, or with OVA-EP-DCs were investigated in vitro. The CFSE-labeled OT-1 CD8^+^ T cells (5 × 10^5^) were cultured for 5 days in each group. CFSE fluorescence intensity of OT-1 CD8^+^ T cells was measured by flow cytometry. The representative histograms of CFSE intensity and the proportion of CFSE^low^ OT-1 cells are shown (*n* = 3). To analyze statistical differences among multiple groups, ANOVA with Holm’s post hoc test was performed. ** *p* <0.01, *** *p* < 0.001, and **** *p* < 0.0001.

**Figure 2 cancers-14-00171-f002:**
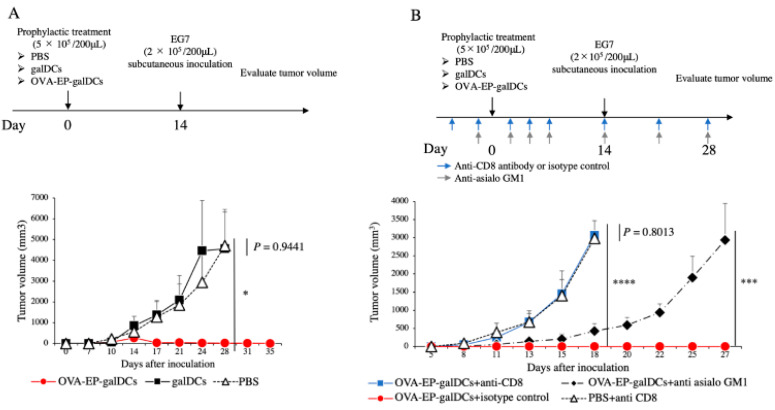
OVA-EP-galDC vaccine promoted complete rejection of subcutaneous tumors in a manner dependent on CD8^+^ T cells in prophylactic models. OVA-electroporated cells were sequentially incubated with α-GalCer and LPS, and adjusted as OVA-electroporated vaccine vectors (OVA-EP-galDCs). Similarly, galDCs without electroporation and PBS were prepared for the control. (**A**) Female C57BL/6 mice were prophylactically treated with PBS, galDCs, or OVA-EP-galDCs (5 × 10^5^ cells/mouse) intravenously (day 0). On day 14 after the treatment, 2 × 10^5^ EG7 were inoculated subcutaneously (*n* = 5). The volume of tumor was calculated by the following formula [V = 1/2 (L × W^2^)], where L = length and W = width. (**B**) The tumor challenge model with CD8^+^ T cell or NK cell-depleted mice. C57BL/6 mice were treated intravenously with OVA-EP-galDCs or PBS on day 0 and were intraperitoneally injected mouse anti-CD8 monoclonal antibody or isotype control on day −3, −1, 2, and 5, and then every 7 days or anti-asialo GM1 antibody on day −1, 2, and 5, and then every 7 days until day 28. Fourteen days after the intravenous treatment, 2 × 10^5^ EG7 were inoculated subcutaneously (*n* = 5). The Student’s t-test was performed to analyze the differences in tumor volume between the two groups. * *p* < 0.05, *** *p* < 0.001, and **** *p* < 0.0001.

**Figure 3 cancers-14-00171-f003:**
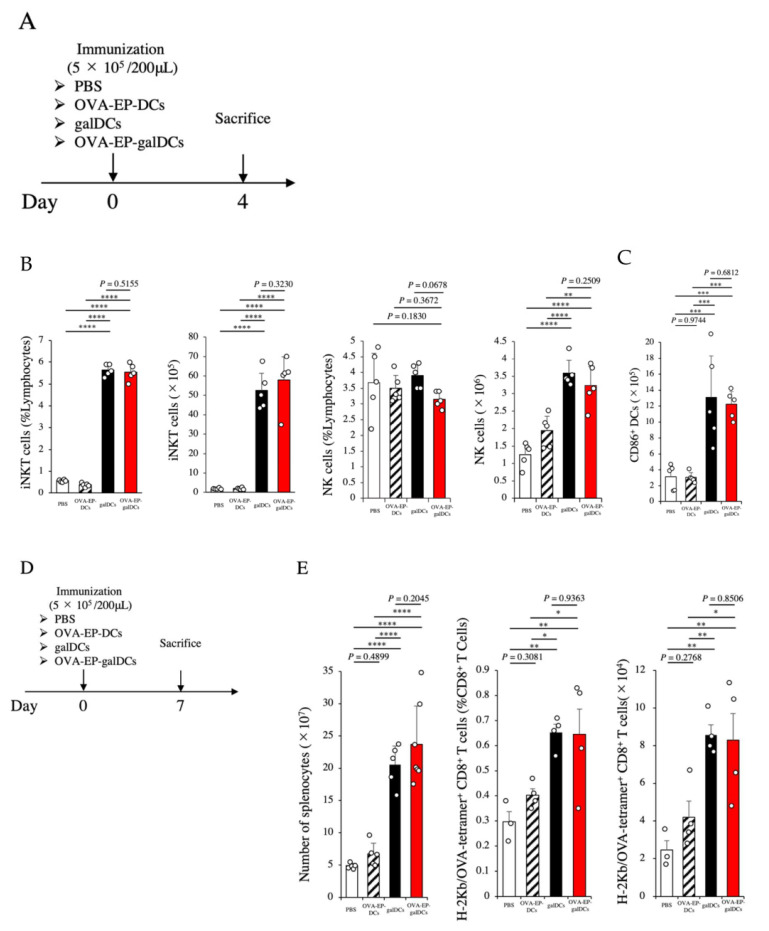
OVA-EP-galDCs induced iNKT cells, NK cells, and DCs in the spleen. (**A**) Experimental schema is shown. C57BL/6 mice were immunized with PBS, OVA-EP-DCs, galDCs, or OVA-EP-galDCs (5 × 10^5^ cells/mouse). OVA-EP-DCs mean OVA-electroporated DCs stimulated with LPS but not pulsed with α-GalCer. On day 4 after the immunization, the frequency and number of iNKT cells and NK cells, and the maturation of the DCs in the spleen were assessed by flow cytometry. (**B**) The frequency and the number of iNKT cells and NK cells in the spleen (*n* = 5). The iNKT cells were defined as CD4^+^ CD1d tetramer^+^ cells and NK cells were defined as CD3^−^ NK1.1^+^ cells. (**C**) The expression of CD86, which is the DC activation marker on splenic DCs, was assessed by flow cytometry. The DCs were defined as CD11c^+^ cells. The number of CD86^+^ DCs is shown in each group (*n* = 5). (**D**) Experimental schema is shown. C57BL/6 mice were immunized with PBS, OVA-EP-DCs, galDCs, or OVA-EP-galDCs (5 × 10^5^ cells/mouse). On day 7 after the immunization, the number of splenocytes was counted. (**E**) The number of total splenocytes and the frequency and the number of OVA-specific CD8^+^ T cells in the spleen were assessed by flow cytometry (*n* = 5). OVA-specific CD8^+^ T cells are defined as H-2Kb/OVA-tetramer^+^ CD8^+^ cells in this figure. To analyze statistical differences among multiple groups, ANOVA with Holm’s post hoc test was performed. * *p* < 0.05, ** *p* < 0.01, *** *p* < 0.001, and **** *p* < 0.0001.

**Figure 4 cancers-14-00171-f004:**
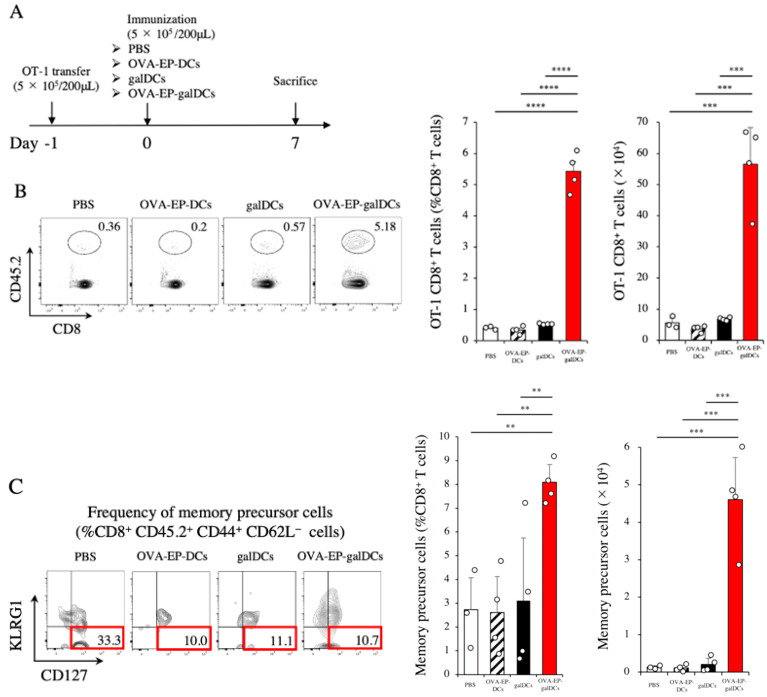
OVA-EP-galDCs activated OT-1 CD8^+^ effector T cells in the spleen. (**A**) C57BL/6-background CD45.1^+^ congenic mice were transferred with 5 × 10^5^ CD45.2^+^ OT-1 CD8^+^ T cells (day −1), and one day later were immunized with PBS, OVA-EP-DCs, galDCs, or OVA-EP-galDCs (5 × 10^5^ cells/mouse). On day 7, OT-1 CD8^+^ T cells (CD45.2^+^ CD8^+^ T cells) and memory precursor cells (CD44^+^ CD62L^−^ KLRG1^−^ CD127^+^ CD45.2^+^ CD8^+^ T cells) in the spleen were assessed by flow cytometry. (**B**) Typical examples of flow cytometric analysis, the frequency, and the number of OT-1 CD8^+^ T cells in the spleen are shown (*n* = 3,4). (**C**) Typical examples of flow cytometric analysis, the frequency, and the number of memory precursor cells in the spleen are shown (*n* = 3,4). To analyze statistical differences among multiple groups, ANOVA with Holm’s post hoc test was performed. ** *p* < 0.01, *** *p* < 0.001, and **** *p* < 0.0001.

**Figure 5 cancers-14-00171-f005:**
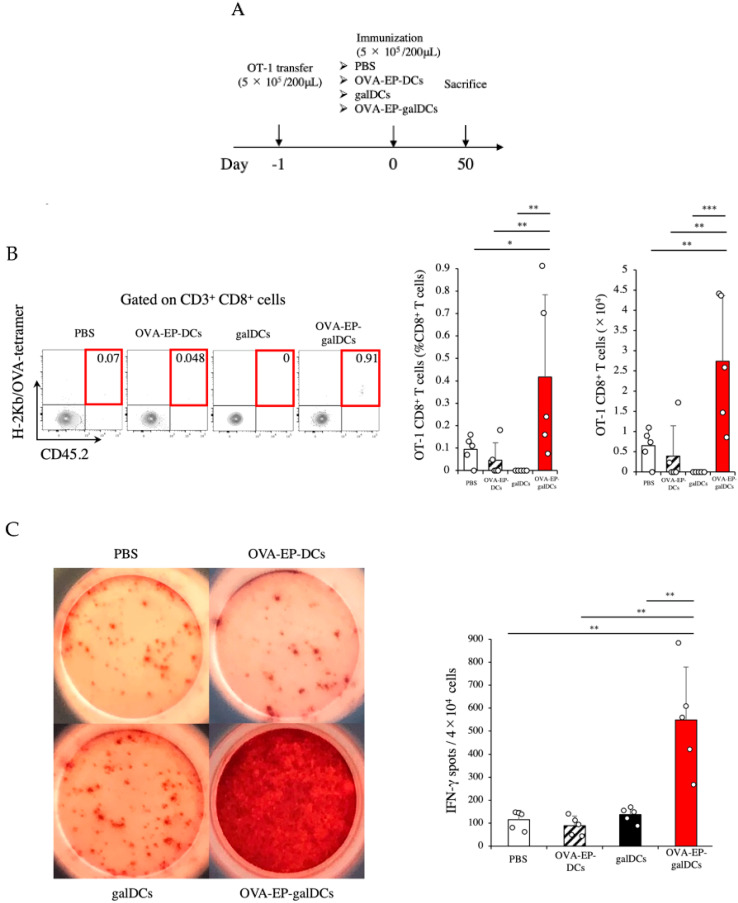
OVA-EP-galDCs induced OT-1 CD8^+^ memory T cells in the spleen. (**A**) On day 50, the frequency and the number of OT-1 CD8^+^ T cells (H-2Kb/OVA-tetrameter^+^ CD45.2^+^ CD8^+^ T cells) in the spleen were assessed by flow cytometry. (**B**) Typical examples of flow cytometric analysis, the frequency, and the number of OT-1 CD8^+^ T cells in the spleen are shown (*n* = 5). (**C**) IFN-γ secretion of splenic CD8^+^ T cells was assessed by ELISPOT assay on day 50. Splenic CD8^+^ T cells were sorted by MACS and cocultured with DCs stimulated with OVA peptide (SIINFEKL) and the IFN-γ spots were shown. The number of spot-forming cells was counted manually (*n* = 5). To analyze statistical differences among multiple groups, ANOVA with Holm’s post hoc test was performed. * *p* < 0.05, ** *p* < 0.01, and *** *p* < 0.001.

**Figure 6 cancers-14-00171-f006:**
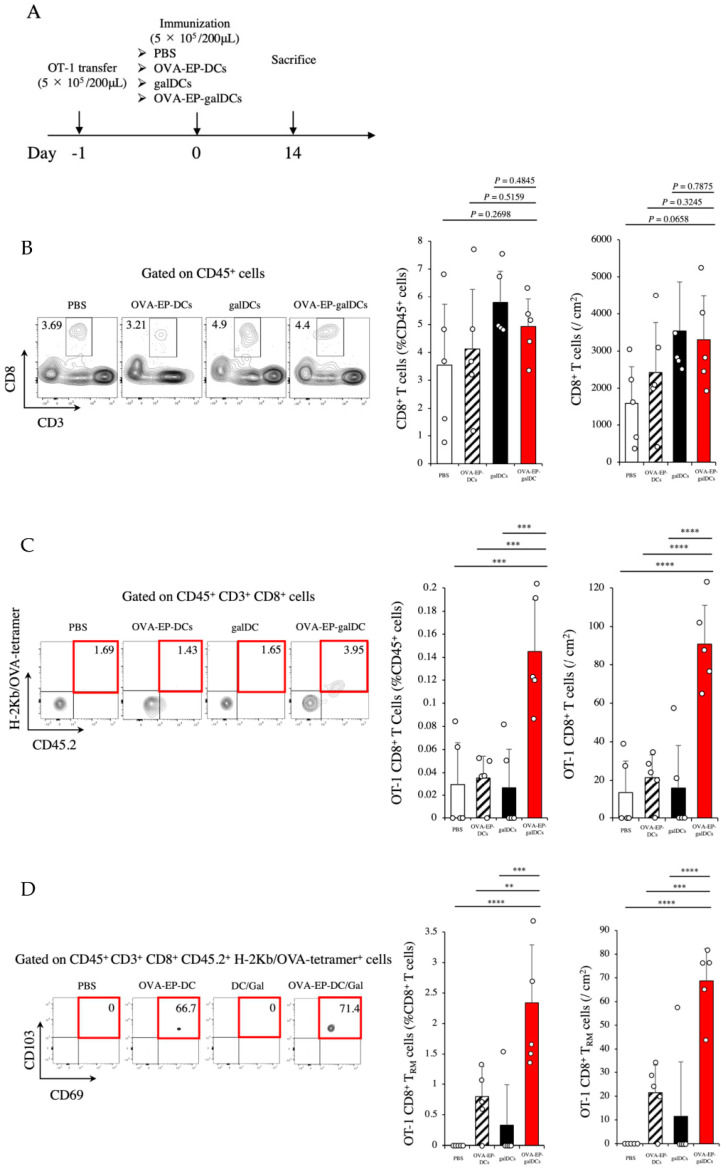
OVA-EP-galDCs induced OT-1 CD8^+^ tissue-resident memory T cells (T_RM_ cells) in the skin. (**A**) Experimental schema is shown. C57BL/6 mice were immunized with PBS, OVA-EP-DCs, galDCs, or OVA-EP-galDCs (5 × 10^5^ cells/mouse). (**B**–**D**) On day 14 after immunization, the frequency and the number of (**B**) CD8^+^ T cells (CD45^+^ CD3^+^ CD8^+^ T cells), (**C**) OT-1 CD8^+^ T cells (H-2Kb/OVA-tetrameter^+^ CD45.2^+^ CD3^+^ CD8^+^ T cells), and (**D**) tissue-resident memory OT-1 CD8^+^ T cells (H-2Kb/OVA-tetrameter^+^ CD45.2^+^ CD3^+^ CD8^+^ CD69^+^ CD103^+^ T cells; OT-1 CD8^+^ T_RM_ cells) in the skin were assessed by flow cytometry. To analyze statistical differences among multiple groups, ANOVA with Holm’s post hoc test was performed. ** *p* < 0.01, *** *p* < 0.001, and **** *p* < 0.0001.

## Data Availability

The data presented in this study are available on request from the corresponding author. The data will be kept for at least 15 years after this publication at a secure location at Kobe University Hospital in Hyogo, Japan.

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
