# Peer review of "Vaccine Based on Dendritic Cells Electroporated with an Exogenous Ovalbumin Protein and Pulsed with Invariant Natural Killer T Cell Ligands Effectively Induces Antigen-Specific Antitumor Immunity"

_cancers, 2021, doi:10.3390/cancers14010171_

Round 1

Reviewer 1 Report

The data described in this manuscript are clearly presented and of relevance for the potential development of model vaccines in cancer. However, the conclusions reached must be put into context due to the use of only one model antigen, ovalbumin (OVA). This is not a tumor-associated antigen but an exogenous one, therefore there are no tolerant T cells to this antigen in the host mouse. Contrarily, most tumor-associated antigens generate some degree of tolerance, thereby the difficulty of raising T-cell-based immunity against them. In addition, the OVA-EG7 tumor model used in this work might highly express OVA, further increasing the T cell response to the tumor. Moreover, the amount of OVA introduced into dendritic cells by electroporation is probably higher than any antigen present in a tumor’s whole cell lysate. All these conditions increase the T cell response against OVA beyond what would be expected from a more physiological situation.

Considering the comments above, it is important that the manuscript’s title reveals the work done in the manuscript. Indeed, the sentence “Vaccine based on dendritic cells electroporated with whole protein…”  is misinforming. The type of antigen used throughout this work (OVA) should be revealed, for example by saying, “a model antigen”, “ovalbumin”, “an exogenous protein”…  Otherwise, the term “whole protein” prompts the reader to think that a “whole-cell extract” has been used.

Other points to be considered:

Line 26           The same comment above on the nature of the used antigen applies to the sentence in the Abstract: “bone marrow-derived dendritic cells are electroporated with tumor-associated whole proteins and…” This sentence is not true, since the dendritic cells are electroporated with ovalbumin, and this should be stated.

Line 329         Regarding the sentence, “As a result, the OVA-EP-galDC vaccine prolonged survival of the tumor-bearing mice compared with controls”. It seems inaccurate since the control mice galDC attain a survival similar to that of the sample OVA-EP-galDC.

Figure 4C

The numbers inside the dot-plots (red squares) do not seem to match with the graph of the frequency of CD8+ T cells. The percentage of IL-7R-positive T cells in PBS control is 33% (dot-plot), whereas in the corresponding graph the percentage of CD8+ T cells is ~3%. However, the percentage of IL-7R-positive T cells in OVA-EP-galDC (dot-plot) is 10.7%, but in the graph, the percentage of CD8+ T cells is ~8%. Please revise.

MINOR POINTS

Line 53           It is unclear what the authors want to say in the sentence, " There are various methods to load tumor antigen information to the hosts,".

Please revise.

Line 76           In the sentence, “iNKT cells interacts CD8α+ DCs.” Maybe the authors wanted to say: “iNKT cells interact with CD8α+ DCs.”

Line 166         Is this sentence the title of a subsection? “Electroporation and coculture of imDCs with proteins”. Please revise.

Lines 171 and 177    The concentration of OVA used is mentioned. However, it is unclear to the reader whether the final OVA concentration when the cells are incubated with OVA but not electroporated (20µg/mL) is the same as the OVA concentration when cells are electroporated.

Line 187         The meaning of the last part of the sentence is unclear. "imDCs were cultured with 100ng/ml of α-GalCer for 40 hours before cell harvesting for α-GalCer pulsation" Maybe the authors wanted to say: "imDCs were cultured with 100ng/ml of α-GalCer for 40 hours before cell harvesting".

Line 385         In the sentence, “C57BL/6-background CD45.1+ congenic mice were transferred with 5 × 105 CD45.2+ OT-1 CD8+ T cells one day later (Day -1), and immunized with PBS...”. Maybe the authors wanted to say:

“C57BL/6-background CD45.1+ congenic mice were transferred with 5 × 105 CD45.2+ OT-1 CD8+ T cells (Day -1), and one day later were immunized with PBS...”

References

I am not sure whether reference 32 is cited in the main text.

Author Response

December 27, 2021

Response letter to REVIEWER #1

We would like to thank the reviewers for their thoughtful comments and for giving us the opportunity to revise the manuscript. We have addressed the reviewers' comments point by point as follows and revised the manuscript accordingly.

Comment 1: it is important that the manuscript’s title reveals the work done in the manuscript. Indeed, the sentence “Vaccine based on dendritic cells electroporated with whole protein…”  is misinforming. The type of antigen used throughout this work (OVA) should be revealed, for example by saying, “a model antigen”, “ovalbumin”, “an exogenous protein”…  Otherwise, the term “whole protein” prompts the reader to think that a “whole-cell extract” has been used.

Response 1: As we totally agreed with the reviewer’s comment, we corrected the title as “Vaccine based on dendritic cells electroporated with an exogenous ovalbumin protein and pulsed with invariant natural killer T cell ligands effectively induces antigen-specific antitumor immunity.” Moreover, to avoid misleading the readers, we avoided "whole" text and used "full-length" text as much as possible (Lines 37, 97, 261, 294, 452, 462, 463, and 532).

Comment 2: Regarding Line 26, the same comment above on the nature of the used antigen applies to the sentence in the Abstract: “bone marrow-derived dendritic cells are electroporated with tumor-associated whole proteins and…” This sentence is not true since the dendritic cells are electroporated with ovalbumin, and this should be stated.

Response 2: According to the reviewer’s comment, we corrected the corresponding term to “an exogenous ovalbumin protein” or “an exogenous protein” (Tilte, citation, lines 26, 35, and 95).

Comment 3: Line 329.  Regarding the sentence “As a result, the OVA-EP-galDC vaccine prolonged survival of the tumor-bearing mice compared with controls,” it seems inaccurate since the control mice galDC attain a survival similar to that of the sample OVA-EP-galDC.

Response 3: In the original manuscript, the term "control" meant the treatment with PBS alone but did not include the treatment with α-GalCer, which led to the above misunderstanding. Based on the reviewer’s comment, we replaced the term "controls" to the term "PBS treatment" in the revised manuscript to specify each group.

Comment 4: Regarding Figure 4C, the numbers inside the dot plots (red squares) do not seem to match with the graph of the frequency of CD8+ T cells. The percentage of IL-7R-positive T cells in PBS control is 33% (dot-plot), whereas in the corresponding graph the percentage of CD8+ T cells is ~3%. However, the percentage of IL-7R-positive T cells in OVA-EP-galDC (dot-plot) is 10.7%, but in the graph, the percentage of CD8+ T cells is ~8%. Please revise.

Response 4: As we totally agreed with the reviewer’s comment, we changed the graph title of the Figure 4C upper panels to “Frequency of memory precursor cells (%CD8+ CD45.2+ CD44+CD62L- cells)” to clarify the parent groups. The Y-axes of the Figure 4C lower panels were left “Memory precursor cells (%CD8+ T cells)” as it is.

Comment 5: Regarding Line 53, it is unclear what the authors want to say in the sentence, " There are various methods to load tumor antigen information to the hosts,".

Response 5: According to the reviewer’s comment, we revised the corresponding statements as follows for better understandability:

“Various methods are applied for loading tumor antigen information to the hosts, including the use of tumor tissues as well as tumor-associated proteins, peptides, and nucleic acid information” (Lines 54-56)

Comment 6: Line 76.  In the sentence, “iNKT cells interacts CD8α+ DCs.” Maybe the authors wanted to say: “iNKT cells interact with CD8α+ DCs.”

Response 6: We corrected the grammatical error the reviewer pointed out as suggested.

Comment 7: Line 166. Is this sentence the title of a subsection? “Electroporation and coculture of imDCs with proteins”. Please revise.

Response 7: We corrected the formatting error the reviewer pointed out.

Comment 8: Regarding Lines 171 and 177, the concentration of OVA used is mentioned. However, it is unclear to the readers whether the final OVA concentration when the cells are incubated with OVA but not electroporated (20 µg/mL) is the same as the OVA concentration when cells are electroporated.

Response 8: Unfortunately, we have no data regarding the OVA signal intensity when the imDCs were electroporated with 20 µg/mL of OVA. Instead, we tested to co-culture the imDCs with 100 μg/ml of OVA. As shown below (Panel A), the imDCs co-cultured with 100 μg/ml of OVA produced almost identical signal intensities to the imDCs electroporated with the same concentration of OVA. Based on these findings, we speculate that, if the administration dose of OVA is the same, the signal intensities of OVA would be almost identical between these two groups. However, as shown below (Panel B), the frequency of cell death was much higher in the imDCs co-cultured with 100 μg/ml of OVA. We assume that DC exhaustion due to excessive external antigen phagocytosis led to the higher frequency of cell death in this group. To maximize the dose of OVA to administer as well as to minimize the treatment-induced cell death, we chose the OVA concentration of 20 μg/ml for co-culturing and 100 μg/ml for electroporation.

Based on the discussion above, we revised the Results Section 3.1 as follows:

“In parallel, various concentrations of OVA were tested in a dose-escalation manner to determine the appropriate OVA concentration for coculture. The imDCs cocultured with 100 μg/ml of OVA produced almost identical signal intensities to the imDCs electroporated with the same concentration of OVA but died at higher frequencies (more than 80%; data not shown). To maximize the dose of OVA to administer as well as to minimize the treatment-induced cell death, we chose the OVA concentration of 20 μg/ml for coculturing and 100 μg/ml for electroporation.” (Lines 273-279)

Comment 9: Regarding Line 187, the meaning of the last part of the sentence is unclear. “imDCs were cultured with 100 ng/ml of α-GalCer for 40 hours before cell harvesting for α-GalCer pulsation” Maybe the authors wanted to say: “imDCs were cultured with 100 ng/ml of α-GalCer for 40 hours before cell harvesting”.

Response 9: We really appreciate the reviewer for generous assist to gain better readability of the manuscript. We revised the corresponding description exactly as suggested.

Comment 10: Line 385. In the sentence, “C57BL/6-background CD45.1+ congenic mice were transferred with 5 × 105 CD45.2+ OT-1 CD8+ T cells one day later (Day -1), and immunized with PBS...”. Maybe the authors wanted to say: “C57BL/6-background CD45.1+congenic mice were transferred with 5 × 105 CD45.2+ OT-1 CD8+ T cells (Day -1), and one day later were immunized with PBS...”

Response 10: We revised the corresponding description as the reviewer suggested.

Comment 11: I am not sure whether reference 32 is cited in the main text.

Response 11: We thank the reviewer for notifying us of the missing citation. Reference 32 has been cited in the Section 2.8.

Thank you for your consideration. We look forward to hearing from you.

Kimihiro Yamashita

Division of Gastrointestinal Surgery, Department of Surgery

Graduate School of Medicine, Kobe University

7-5-2, Kusunoki-cho, Chuo-ku, Kobe, Hyogo, 650-0017, Japan

Tel: 81-78-382-5925 / Fax: 81-78-382-5939

Reviewer 2 Report

The authors have developed a new cancer vaccine strategy using OVA protein-loaded dendritic cells by electroporation and pulsed with α-GalCer (OVA-EP-galDCs). The OVA-EP-galDCs were intravenously administered to mice as a vaccine can inhibit tumor growth. The antitumor effects are mediated by CD8+ T cells. In addition, the authors showed that immunization with OVA-EP-galDCs can induce antigen-specific TRM cells.

In general, the manuscript is well written and well present for reader. They proposed the design for cancer immunotherapy. It is worth to published in this journal.

However, in its present form, the paper reports must address some concerns that are listed below.

Minor concerns:

  1. The absence of therapeutic treatment results indicated that this approach needs to be improved. Please discuss how to improve the therapeutic effects in future.
  2. The anti-asialo GM1 depletion may have some role in the antitumor mechanisms (Fig. 2B). Please discuss this part.
  3. The antitumor effects of OVA-EP-DCs need to be included or described in this paper.
  4. Line 186, “100 of ng/ml” should be “100 ng/ml “.

Author Response

December 27, 2021

Response letter to REVIEWER #2

We would like to thank the reviewers for their thoughtful comments and for giving us the opportunity to revise the manuscript. We have addressed the reviewers' comments point by point as follows and revised the manuscript accordingly.

Comment 1: The absence of therapeutic treatment results indicated that this approach needs to be improved. Please discuss how to improve the therapeutic effects in future.

Response 1: As the reviewer pointed out, the OVA-EP-galDC vaccine did not show superior therapeutic effects to the galDC vaccine in the lung metastasis model (Supplementary Fig. S2). These results may reflect a peculiarity of the model in which NK cells play more important roles than T cells in the control of lung metastases as we have already discussed.

“Although the OVA-EP-galDC vaccine successfully rejected subcutaneous tumors, the galDC vaccine alone had little effects on the tumor growth (Figure 2A) even with the activation of NK cells in both vaccines (Figure 3C). (…omitted…) As the effector functions of NK cells, neither CD4+ nor CD8+ T cells, have been found to be important in the control of metastatic lung tumors [52, 53], we tested our DC vaccine system in a mouse lung metastasis model (Supplementary Figure S2). As a result, the galDC vaccine alone well controlled the growth of lung metastases in mice, suggesting the antitumor effector functions of NK cells.” (Lines 499–508)

Regarding how to improve the therapeutic effects of the OVA-EP-galDC vaccine, we have added the following statement in the Discussion section.

“The combination therapy of α-GalCer and immune checkpoint inhibitors (ICIs) has been shown to have synergistic effects [54]. To improve the therapeutic effects of the OVA-EP-galDCs, we are currently addressing to combine this system with ICIs as not only a prophylactic strategy but a therapeutic strategy for established tumors.” (Lines 5­17–520)

Comment 2: The anti-asialo GM1 depletion may have some role in the antitumor mechanisms (Fig. 2B). Please discuss this part.

Response 2: As we agreed to the reviewer that the roles of NK cells in the presence of antigen-specific T cells should be discussed, we added the following statements in the Discussion section.

“In addition, NK cell depletion by the anti-asialo GM1 antibody after the OVA-EP-galDC vaccine reduced the therapeutic effects (Figure 2B). These results suggest that NK cells play at least a supportive/modulative role in the effector functions of antigen-specific CD8+ T cells.” (Lines 5­08–511)

Comment 3: The antitumor effects of OVA-EP-DCs need to be included or described in this paper.

Response 3: As mentioned in the response 1, the lung metastasis model is not applicable to the above research question since NK cells are more important than antigen-specific T cells in this model. The effects of the OVA-EP-DCs in the prophylactic model have not yet been evaluated and need to be verified in the future. However, the OVA-EP-DCs were insufficient in inducing antigen-specific skin TRM cells, which reflect the therapeutic effects (Figure 3D). Therefore, we assume that the effects of the OVA-EP-DCs alone would be weak. In contrast, in the established tumor model, the OVA-EP-DCs (as well as the OVA-EP-galDCs) are expected to be effective in combination with ICIs, and this point should be verified in the future.

Based on the discussion above, we have added the following statements in the Discussion section.

“It is also important to note that ICIs enhance the effector functions of antigen-specific T cells. In this study, however, we did not fully evaluate the therapeutic effects of the OVA-EP-DCs that are expected to induce antigen-specific T cells (Figure 2) because we primarily focused on the significance of α-GalCer-mediated immune responses in the EP-DC system. In this regard, the OVA-EP-DCs were insufficient to induce antigen-specific skin TRM cells, which reflect the therapeutic effects (Figure 3D). Therefore, the effects of the OVA-EP-DCs are presumed to be weak. Nevertheless, particularly in the established tumor models, the OVA-EP-DCs (as well as the OVA-EP-galDCs) are expected to be effective in combination with ICIs, and this point should be verified in the future.” (Lines 5­20–529)

Comment 4: Line 186, “100 of ng/ml” should be “100 ng/ml “.

Response 4: According to the reviewer’s comments, we revised the corresponding term “100 of ng/ml” to “100 ng/ml “.

Thank you for your consideration. We look forward to hearing from you.

Kimihiro Yamashita

Division of Gastrointestinal Surgery, Department of Surgery

Graduate School of Medicine, Kobe University

7-5-2, Kusunoki-cho, Chuo-ku, Kobe, Hyogo, 650-0017, Japan

Tel: 81-78-382-5925 / Fax: 81-78-382-5939
